# Drug Discovery Using Evolutionary Similarities in Chemical Binding to Inhibit Patient-Derived Hepatocellular Carcinoma

**DOI:** 10.3390/ijms23147971

**Published:** 2022-07-19

**Authors:** Jin Hong Lim, Keunwan Park, Kyung Hwa Choi, Chan Wung Kim, Jae Ha Lee, Raymond Weicker, Cheol-Ho Pan, Seok-Mo Kim, Ki Cheong Park

**Affiliations:** 1Gangnam Severance Hospital, Department of Surgery, Yonsei University College of Medicine, 211 Eonjuro, Gangnam-gu, Seoul 06273, Korea; doctorjin@yuhs.ac; 2Natural Product Informatics Research Center, KIST Gangneung Institute of Natural Products, Gangneung 25451, Korea; keunwan@kist.re.kr (K.P.); panc@kist.re.kr (C.-H.P.); 3Department of Urology, CHA Bundang Medical Center, CHA University, Seongnam 13496, Korea; khchoi@cha.ac.kr; 4CKP Therapeutics, Inc., 110 Canal Street, 4th Floor, Lowell, MA 01852, USA; ckim@ckptherapeutics.com (C.W.K.); jlee@ckptherapeutics.com (J.H.L.); rweicker@ckptherapeutics.com (R.W.); 5Department of Surgery, Yonsei University College of Medicine, 50-1, Yonsei-ro, Seodaemun-gu, Seoul 03722, Korea

**Keywords:** patient-derived anti-cancer drug-resistant hepatocellular carcinoma, endoplasmic reticulum stress, sarcoplasmic/endoplasmic reticulum calcium ATPase, thapsigargin, candidate 19, candidate 23, cancer stem cells

## Abstract

Drug resistance causes therapeutic failure in refractory cancer. Cancer drug resistance stems from various factors, such as patient heterogeneity and genetic alterations in somatic cancer cells, including those from identical tissues. Generally, resistance is intrinsic for cancers; however, cancer resistance becomes common owing to an increased drug treatment. Unfortunately, overcoming this issue is not yet possible. The present study aimed to evaluate a clinical approach using candidate compounds 19 and 23, which are sarcoplasmic/endoplasmic reticulum calcium ATPase (SERCA) inhibitors, discovered using the evolutionary chemical binding similarity method. mRNA sequencing indicated SERCA as the dominant marker of patient-derived anti-cancer drug-resistant hepatocellular carcinoma (HCC), but not of patient-derived anti-cancer drug-sensitive HCC. Candidate compounds 19 and 23 led to significant tumor shrinkage in a tumor xenograft model of anti-cancer drug-resistant patient-derived HCC cells. Our results might be clinically significant for the development of novel combinatorial strategies that selectively and efficiently target highly malignant cells such as drug-resistant and cancer stem-like cells.

## 1. Introduction

Independent of presurgical cancer progression, post-surgical recurrence and survival rates have benefitted from systemic presurgical chemotherapies [1,2,3]. However, standard preoperative systemic chemotherapy is not available for hepatocellular carcinoma (HCC). Systemic chemotherapy is recommended for patients with advanced HCC, the symptoms of which include metastatic lesions or portal vein invasion, and localized therapy is not applicable owing to systemic chemotherapy-resistant cancer [4,5]. Metabolic reprogramming of cancer cells has garnered considerable attention as an emerging hallmark of cancer progression [6]. High extent of glycolysis, known as the Warburg effect, is observed in response to an increased need of tumor cells for biosynthetic precursors and can render the cells exquisitely sensitive to glucose deprivation [7,8]. Moreover, an increasing number of malignant cancer cells, such as highly drug-resistant subclones, have acquired a universal survival strategy and are better equipped than nonmalignant cells to cope with evolutionary stresses, including metabolic and genotoxic stresses [9]. Particularly, epigenetic reprogramming of drug-resistant cancer cells is crucial for their survival following treatment of anti-cancer drugs [10,11]. The progression of drug resistance in refractory cancer cells is a critical factor behind the failure of conventional cancer therapy [9,12,13]. This is an extremely acute problem that can lead to cancer recurrence, metastasis, and death [14,15]. The present study aims to elucidate the mechanisms underlying susceptibility of refractory cancer cells to existing therapies. In this study, we investigated the sarcoplasmic/endoplasmic reticulum calcium ATPase (SERCA), a dominant marker of patient-derived anti-cancer drug-resistant HCC but not of patient-derived anti-cancer drug-sensitive HCC. SERCA is crucial in regulating cytosolic free calcium [16]. Moreover, it substantially participates in calcium signaling homeostasis by effectively re-sequestering cytosolic free calcium into the endoplasmic reticulum (ER) once the desired physiological reaction occurs [17,18]. Cytosolic free calcium regulates numerous cellular processes, including cellular life-or-death decisions, programmed cell death, also known as apoptosis, and autophagy [19,20]. We investigated two novel SERCA inhibitors, compounds 19 and 23, as potential therapeutic agents for patient-derived anti-cancer drug-resistant cells.

## 2. Results

### 2.1. Patient Disease Characteristics

In total, 233 patients were consecutively treated with R0 resection between 2016 and 2021. Among them, 37 patients had extrahepatic spread HCC (Figure 1A) and were analyzed in this study. Medical records indicated that before operation, 19 patients did not undergo chemotherapy; one patient underwent chemotherapy with sorafenib, three patients underwent chemotherapy with sorafenib and regorafenib, one patient underwent chemotherapy with gemcitabine and cisplatin, and 13 patients underwent chemotherapy with fluorouracil and cisplatin. The mean age of the patients was 60.2 ± 13.3 y, and 83.8% were males. Disease characteristics and demographics of patients are presented in Figure 1A. Disease-free survival (DFS, Figure 1B) and overall survival (OS, Figure 1C) for the 37 patients were 12.4 ± 12.94, and 20.3 ± 13.80 months, respectively. No significant difference in DFS and OS was observed in patients with or without chemotherapy.

### 2.2. Contrasting Gene Expression and Signaling Stimulation between Patient-Derived Anti-Cancer Drug-Sensitive and Drug-Resistant HCC Cells

To investigate alterations in gene expression and signaling pathways in patient-derived anti-cancer drug-sensitive and drug-resistant HCC, we performed an RNA sequencing (RNA-Seq)-based transcriptome analysis. Diverse patient-derived HCC cells used in this study were detached from the specimens (Figure 2A). YUMC-S-H1 was the first isolated drug-sensitive HCC cell, and YUMC-R-H1, -H2, -H3, -H4, and -H5 were the first, second, third, fourth, and fifth isolated drug-resistant HCC cells, respectively. YUMC-R-H1, -H2, -H3, -H4, and -H5 were more aggressive than YUMC-S-S1. Metastasis and recurrence were observed in patients with drug-resistant HCC cells (Figure 2A). Remarkably, under severe ER stress following anti-cancer drug treatment, these drug-resistant HCC cells exhibited a significantly higher expression of target genes and survival signaling pathways than did the drug-sensitive HCC cells. The divergence of mRNA expression patterns between drug-resistant and -sensitive HCC cells was detected using transcriptome analysis (Figure 2B). Particularly, exposure to genotoxic stressors, such as regorafenib, sorafenib, cisplatin, and gemcitabine, markedly increased the levels of HCC stemness markers (CD13^high^, CD24^high^, CD44^high^, CD90^high^, CD133^high^, EpCAM, ICAM-1, LGR5, MAEL, Cripto-1, and Oct4^high^) and calcium-mediated and survival-related target genes (*ATP2A* isoforms and B-cell lymphoma 2 (*Bcl-2*)) in drug-resistant HCC cells compared with those in drug-sensitive HCC cells (Figure 2B). Figure 2C–G show a comparison of 20 key regulator genes in distinct signaling pathways in drug-resistant and drug-sensitive HCC cells. 

Intriguingly, we discovered that the Notch, calcium, and cancer stemness signaling pathways were highly enriched in drug-resistant HCC cells (Figure 2C–G). Moreover, the levels of SERCA and Bcl-2, key regulator proteins involved in calcium homeostasis and anti-apoptosis, were significantly high in drug-resistant HCC cells (Figure 2H). SERCA is crucial for removing free calcium from the cytoplasm; therefore, we measured the level of *SERCA* isoforms. Expression of *SERCA* isoforms slightly differed between the two types of HCC cells; particularly, mRNA levels of *SERCA1* and *SERCA3* were upregulated in drug-resistant HCC cells (Figure 2I–M). 

These data indicate that the enhancement of gene pathways in regulating stemness and SERCAs is crucial for the survival of drug-resistant HCC cells under anti-cancer drug-treated conditions.

### 2.3. A Novel Therapeutic Trial of Candidates 19 and 23, SERCA Inhibitors, for Patient-Derived Drug-Resistant HCC Treatment via in Silico Screening

Based on the results in Figure 2, we hypothesized that the functional restriction of SERCA in anti-cancer drug-resistant HCC cells could be a possible therapeutic approach. In silico screening for SERCA-binding compounds was used to explore possible pharmacophoric binding modes. Particularly, 1750 (based on binding score), 184 (manual selection), and 78 candidate compounds were identified. Of these, 34 compounds exhibited high binding affinity with SERCA (Figure 3A). The 19th (Figure 3B) and 23rd (Figure 3C) candidate compounds (candidates 19 and 23) significantly inhibited SERCA and were selected for further evaluation as novel therapeutic agents for drug-resistant HCC (Figure 3A–C).

### 2.4. Novel Pharmacophore Candidates 19 and 23 Suppressed the Survival of Drug-Resistant HCC Cells

To assess the anti-cancer effects of candidates 19 and 23, we performed cell viability assays by treating cells with anti-cancer drugs and candidates 19 or 23. The viability of YUMC-S-H1 notably decreased in a dose-dependent manner following treatments of regorafenib, sorafenib, cisplatin, and gemcitabine with or without candidates 19 or 23 (Figure 4A). However, the viabilities of YUMC-R-H1, -H2, -H3, -H4, and -H5 were not significantly affected following anti-cancer drug treatment. Notably, thapsigargin, a well-known SERCA inhibitor (positive control) or candidates 19 or 23 treated in combination with anti-cancer drugs significantly suppressed the viability of drug-resistant HCC cells in a dose-dependent manner (Figure 4B–F, Appendix A). Moreover, treatment with thapsigargin or candidates 19 or 23 alone did not significantly affect the viability of HCC cells. The cytosolic free calcium influx through SERCA and the release of free calcium from the ER into the cytosol is an early and crucial event in apoptosis [21]. A failure to lower the spike of intracellular calcium leads to ER stress and apoptotic cell death [22,23,24]. In drug-sensitive HCC cells, cytosolic free calcium returned to the basal levels after the spike when treated with candidates 19 or 23 alone, whereas treatment with anti-cancer drugs prevented such a return of free calcium levels (Figure 4G). In contrast, free calcium in drug-resistant HCC cells returned to the basal levels when treated with anti-cancer drugs or candidates 19 or 23 alone (Figure 4H–L). Interestingly, combination treatment with anti-cancer drugs and candidate 19 or 23 failed to show similar effects (Figure 4H–L). Half maximal inhibitory concentrations (IC_50_) were 7 μM for regorafenib, 25 μM for sorafenib, 20 μM for cisplatin, and 80 nM for gemcitabine in drug-sensitive HCC cells. In drug-resistant HCC cells, IC_50_ values were 20 μM for regorafenib, 20 μM for sorafenib, 20 μM for cisplatin, and 100 nM for gemcitabine. The doses of candidates 19 (120 μM) and 23 (80 μM) were fixed for single or combined treatment.

These results demonstrated that SERCA is a critical factor in the survival of drug-resistant cancer cells when treated with existing chemotherapies. When existing chemotherapies are used in combination with candidates 19 and 23, cytosolic calcium levels remain elevated and do not return to the basal levels.

### 2.5. SERCA Increased the Anti-Apoptotic Activity of Severe ER Stress- and Drug-Resistant HCC Cells upon Prolonged Anti-Cancer Drug Treatment

The results in Figure 4 show that the increase in SERCA expression might be crucial for cell survival under severe ER stress following anti-cancer drug treatment. Therefore, we performed an immunoblot assay to investigate the relationship between SERCA expression and ER stress. SERCA expression was predominantly induced in drug-resistant (Figure 5E–I) but not in drug-sensitive (Figure 5A–D) HCC cells. Consequently, drug-sensitive HCC cells exhibited a weak response to severe ER stress following drug treatment (Figure 5A–D). In contrast, drug-resistant HCC cells could avoid severe ER stress by inducing SERCA expression. However, treatment of anti-cancer drugs combined with thapsigargin or candidates 19 or 23 markedly increased apoptosis in the sub-G_0_G_1_ phase (Figure 5J–N). To demonstrate that primarily SERCA and not calcium ion channels increased the intracellular/extracellular calcium ratio in drug-resistant HCC cells, we treated the cells with candidates 19 or 23 (novel SERCA inhibitors), nifedipine or verapamil (calcium channel blockers), and regorafenib, sorafenib, cisplatin, or gemcitabine (anti-cancer drugs). Combination treatment with SERCA inhibitors and anti-cancer drugs significantly induced the expression of CHOP, an ER stress marker, in drug-resistant HCC cells, while calcium channel blockers alone or combined with anti-cancer drugs had no significant effect on drug-resistant HCC cells (Figure 5O–S). 

Consequently, these results showed that drug-resistant HCC cells may avoid calcium-mediated apoptosis via ER stress induced by the existing anti-cancer drugs by increased SERCA levels, rather than calcium ion channels.

### 2.6. Novel Therapeutic Trials of Candidates 19 and 23 in Patient-Derived Drug-Resistant HCC Cell-Treated Mouse Xenograft Model

We assessed the anti-cancer effects of candidates 19 and 23 in vivo. We established a mouse xenograft model using patient-derived drug-resistant HCC cells. We administered regorafenib, sorafenib, cisplatin, and gemcitabine to increase genotoxic stress in the mouse xenograft model. Treatment with thapsigargin or candidates 19 or 23 alone was not significantly effective. However, their treatment in combination with anti-cancer drugs significantly induced tumor shrinkage compared to that in the control group and mice treated with each agent alone (Figure 6A–E, left; Appendix A). The increase in spontaneous ER stress upon tumor growth and progression resulted in slight tumor shrinkage following treatment with regorafenib, sorafenib, cisplatin, or gemcitabine, and thapsigargin or candidates 19 or 23 (Figure 6A–E, left; Appendix A). Moreover, the resected tumor weight was markedly lower in the combined treatment group (Figure 6A–E, middle). Treatment with each agent alone or combined treatment did not significantly affect mouse body weight (Figure 5A–E, right). 

## 3. Discussion

Owing to the development of anti-cancer drugs, numerous studies have reported that preoperative chemotherapy improves the survival rates after surgery and demonstrated the effect of properly administered chemotherapy and performed surgery even when effective treatment was considered impossible [25,26,27,28]. Nevertheless, no treatment options have been accepted as the standard neoadjuvant or adjuvant setting for HCC [29,30,31], and considerable numbers of HCC patients with cancer died; therefore, unmet medical needs have steadily increased. 

A significant part of these needs is attributed to drug-resistant recurrent and metastatic cancers [32,33,34,35,36]. Acquisition of drug resistance by refractory cancer cells remains a major obstacle in treating patients with refractory cancers [37,38,39]. Therefore, anti-cancer drug resistance of refractory cancers represents the greatest challenge in cancer therapy [40,41,42]. Numerous underlying mechanisms of drug resistance in patients with refractory cancers have been proposed; each cancer has a distinct set of characteristics required for its progression [43,44]. Therefore, a clinical solution for anti-cancer drug resistance of refractory cancer may appear unattainable. 

Here, we used mRNA-Seq, ECBS, and patient-derived anti-cancer drug-resistant HCC cells to create a framework for drug-resistant cancer therapy. HCC is the second most deadly cancer worldwide and is generally accompanied by chronic hepatic inflammation related to alcohol abuse, viral infection, and metabolic syndromes [44,45]. In recent decades, remarkable progress has been made in HCC treatment [46,47]. Unfortunately, >50% of HCCs progress to an advanced stage, and approximately 70% of patients experience recurrence within the first five years after initiating therapy [48]. Early stage HCC is surgically removed, while advanced HCC frequently requires systemic therapy, including local ablation therapy, irradiation, and trans-arterial chemoembolization [49,50]. However, the effect of systemic chemotherapy on HCC is not as significant as that of surgery. Notably, the relationship between systemic chemotherapy before surgery and improved survival of patients with HCC after surgery has not yet been elucidated. Therefore, preventing the acquisition of anti-cancer drug resistance is important in HCC treatment. 

Previously, we have demonstrated that acute metabolic stress response leads to positive subclone sorting with cancer stem cells (CSCs) [51,52]. Survival signaling pathways are more stimulated in cells with CSC properties than in their progenitors under severe ER stress [53]. Epigenetic alterations in refractory HCC can change the expression of target genes without modifying DNA sequences [54]. In this study, we revealed that the calcium and Notch signaling pathways were among 20 highly enriched signaling pathways in anti-cancer drug-resistant HCC cells compared with those in anti-cancer drug-sensitive HCC cells. 

mRNA-Seq revealed that SERCA was dominantly expressed in anti-cancer drug-resistant HCC cells. Notably, the relationship between the calcium and Notch signaling pathways has been previously reported [55]. Notch signaling is suppressed by SERCA inhibition [55,56]. Therefore, we first focused on crucial genes and the signaling pathways related to cell survival and calcium homeostasis in drug-resistant HCC cells. Intriguingly, we demonstrated that metabolic stress-resistant cancer cells avoided apoptosis mediated by cytosolic free calcium overload via SERCA activation under metabolic stress [52]. However, regardless of the high induction of SERCA expression in drug-resistant HCC cells, functional inhibition of SERCA by thapsigargin or candidates 19 or 23 might lead to apoptosis via an increase in ER stress. Interestingly, despite well-known anti-cancer effects of regorafenib, sorafenib, cisplatin, and gemcitabine, drug-resistant HCC cells were not significantly affected. 

The expression of numerous genes markedly differed between drug-sensitive and drug-resistant HCC cells, suggesting that complex biological processes were reprogrammed in drug-resistant HCC cells to prolong cell survival under severe ER stress caused by anti-cancer drug treatment. Consequently, we hypothesized that the survival of patient-derived anti-cancer drug-resistant HCC cells could lead to the transcriptional stimulation of the calcium and nuclear factor kappa B signaling pathways, which could overburden the cells with cytosolic free calcium and adjust anti-apoptotic and metabolic pathways, respectively, following anti-cancer drug treatment. 

Cytosolic free calcium levels are markedly increased by anti-cancer drug treatment, which causes an increase in calcium-mediated apoptosis of drug-sensitive cancer cells [56,57]. The regulation of cytosolic free calcium signaling represents a critical hallmark procedure for regulating SERCA-induced resistance to apoptosis [55,58]. Moreover, high SERCA levels prevent calcium-mediated apoptosis and are associated with a poor prognosis in patients with cancer [59]. Consequently, we employed the ECBS method to explore the effect of SERCA inhibitors on anti-cancer drug-resistant HCC cells [60]. 

In this study, the inhibition of cytosolic calcium influx into the ER by candidates 19 and 23 caused cytosolic free calcium overload and cell death via severe ER stress. Particularly, genotoxic stress caused by anti-cancer drug treatment increases cytosolic free calcium levels via ER stress [61,62,63]. Notably, the ER is the main intracellular storage site for free calcium and is crucial for regulating calcium homeostasis [64,65,66]. Upon a marked increase in cytosolic free calcium levels, mitochondria buffer and rapidly uptake cytosolic free calcium to maintain calcium homeostasis and avoid free calcium overload [67,68]. However, diminished mitochondrial respiration caused by cytosolic free calcium overload decreases membrane potential and induces cytochrome c release, resulting in apoptosis [69,70,71]. Consequently, the restoration of cytosolic free calcium levels via SERCA could account for the prolonged survival of drug-resistant cancer cells after anti-cancer drug treatment. 

This study is the first to demonstrate that SERCA activation plays a crucial role in preventing apoptosis mediated by cytosolic free calcium overload in patient-derived anti-cancer drug-resistant HCC cells. Therefore, SERCA activation in drug-resistant HCC cells might represent one of the most effective mechanisms of prolonged HCC cell survival. We showed that anti-cancer drug-resistant HCC cells could be killed by selectively inhibiting SERCA under severe ER stress following anti-cancer drug treatment. No evidence of systemic toxicity or treatment-related death was observed in any of the groups.

## 4. Materials and Methods

### 4.1. Study Design and Ethical Considerations

This study was a single center examination of patients diagnosed with HCC over a 7-year period (January 2015–December 2021). A chart of the study design is shown in Appendix A. Cancer cells were obtained from resected specimens of the patients at the Gangnam Severance Hospital, Yonsei University College of Medicine, Seoul, Korea. All patient procedures and courses of treatments complied with institutional ethical standards, national/local regulations, and guidelines of the 1964 Helsinki Declaration as amended. Study methods were authorized by the Institutional Review Board (IRB) of Severance Hospital, Yonsei University College of Medicine (IRB protocol: 3-2019-0281).

### 4.2. Patient Characteristics

Patient-derived hepatocellular carcinoma (HCC), YUMC-S-H1, -R-H1, -R-H2, -R-H3, -R-H4, and -R-H5 C1 cells were obtained from tumor specimens of patients following their last surgery (Table 1). 

#### 4.2.1. Patient for Sample 1

Patient was a 63-year-old male with HCC and the patient had hepatitis B virus (HBV)-related liver cirrhosis, with a 2.5 cm mass observed in the left lateral sector of the liver during a regular health check-up. The patient underwent laparoscopic left lateral sectionectomy. Specimens for culture were collect after the surgery. The pathology state revealed HCC. 

#### 4.2.2. Patient for Samples 2 and 6

Patient was a 53-year-old woman with HCC. She had jaundice with perihilar mass and Klatskin type IIIb bile duct obstruction. She underwent left trisectionectomy and caudate segmentectomy with radical resection of bile duct in October 2019. Pathology state reveals HCC. One month post-surgery, abdominal carcinomatosis was confirmed on abdominal and pelvic computed tomography (APCT) for which she received two cycles of sorafenib. The disease had a partial response. However, the course of treatment was changed due to the progression of her cancer and chemotherapy induced complications (hand–foot syndrome). She underwent palliative abdominal wall mass excision for a metastatic lesion that had penetrated the skin; a fresh specimen for culture was obtained post-surgery. Following this, the chemotherapy regimen was changed to regorafenib, and stopped after three cycles of regorafenib due to chemotherapy complication. A subsistent tumor was later located in the xiphoid area and she went under surgical excision from which the specimens for culture were obtained. 

#### 4.2.3. Patient for Samples 3 and 4

Patient was a 70-year-old male with HBV-related early liver cirrhosis. High levels of alpha-fetoprotein were discovered during an examination and he was referred to Gangnam Severance Hospital in February 2019. His APCT revealed a 3.3 cm single arterial enhancing lesion in the right lobe of liver. He underwent laparoscopic right anterior sectionectomy in March 2019. A follow-up outpatient post-surgical examination and APCT identified a 2.1 cm enlarged aortocaval lymph node. Laparoscopic aortocarval lymph node dissection was performed in May 2019 and the pathology state revealed metastatic carcinoma. After the surgery in May, the patient underwent two cycles of sorafenib after which the disease progression was confirmed in the anti-cancer drug response evaluation. The chemotherapy regimen was changed to regorafenib. After four cycles of regorafenib, the disease did not have a response. Tumors were observed in the celiac trunk lymph node and superior mesenteric lymph node. The patient requested surgery because of the side effects of chemotherapy. Multiple lymph node dissections were performed and the fresh specimen for culture was obtained after surgery. Tumor recurrence continued even after six surgeries; palliative chemotherapy with fluorouracil and cisplatin (FP) was prescribed. After the 6th FP chemotherapy cycle, the tumor was localized to the retropancreatic area. A tumorectomy was performed and a specimen for culture was obtained after surgery.

#### 4.2.4. Patient for Sample 5

Patient was a 64-year-old male with HCC. He underwent segment 5 wedge resection of liver in June 2015. Three years after surgery, metastasis to abdominal wall was discovered, and wide excision of abdominal wall mass was performed. Post-surgery, gemcitabine and cisplatin chemotherapy was prescribed, after 6 cycles the tumor recurred in right lobe of liver. Right hepatectomy was performed in March 2019. The specimens for culture were collected after surgery.

### 4.3. Patient Tissue Specimens

Fresh tumor samples were collected from patients who underwent surgical resection and had biochemically and histologically established drug resistance, metastatic and recurrent cancer with metastasis, and were treated at the Severance Hospital, Yonsei University College of Medicine, Seoul, Korea. 

### 4.4. Ethical Considerations

The research protocol was approved by the Institutional Review Board of Severance Hospital, Yonsei University College of Medicine (Institutional Review Board Protocol: 3-2019-0281). Cell samples were obtained from patients at the Severance Hospital, Yonsei University College of Medicine, Seoul, South Korea.

### 4.5. Tumor Cell Isolation and Primary Culture

After resection, tumor tissue samples were maintained in normal saline supplemented with anti-fungal and antibiotic agents and transferred to the laboratory. Normal tissue and fat were removed and the tumor tissues were rinsed with 1× Hank’s balanced salt solution. Further protocol details are as published [72].

### 4.6. mRNA-Seq Data

We preprocessed the raw reads from the sequencer to remove low-quality and adapter sequences before analysis and aligned the processed reads to Homo sapiens (GRCh37) using HISAT v2.1.0 (KIM et al., 2015) [73]. HISAT utilizes two types of indexes for alignment: a global, whole-genome index, and tens of thousands of small local indexes. Both are constructed using the same Burrows–Wheeler transform (BWT) or graph FM index (GFM) as Bowtie2. Due to the use of these efficient data structures and algorithms, HISAT generates spliced alignments several times faster than Bowtie and the widely used BWA. The reference genome sequence of Homo sapiens (GRCh37) and annotation data were downloaded from the National Center for Biotechnology Information (NCBI). Then, transcript assembly of known transcripts was processed using StringTie v2.1.3b (Pertea, Mihaela et al., 2015, 2016). Based on these results, expression abundances of transcripts and genes were calculated as read count or fragments per kilobase of exon per million fragments mapped (FPKM) value per sample. The expression profiles were used for additional analyses, such as of differentially expressed genes (DEGs). In groups with different conditions, differentially expressed genes or transcripts were filtered through statistical hypothesis testing. Further protocol details are as previously published [74].

### 4.7. Statistical Analysis of Gene Expression Level

The relative abundances of genes were measured in read count using StringTie. We performed statistical analyses to find differentially expressed genes using the estimates of abundances for each gene in the samples. Genes with one more than zero read count values in the samples were excluded. To facilitate log2 transformation, 1 was added to each read count value of filtered genes. Filtered data were log2-transformed and subjected to trimmed mean of M-values (TMM) normalization. The statistical significance of the differential expression data was determined using exactTest, edgeR, and fold change, in which the null hypothesis was that no difference exists among groups. False discovery rate (FDR) was controlled by adjusting the *p*-value using the Benjamini–Hochberg algorithm. For DEG sets, hierarchical clustering analysis was performed using complete linkage and Euclidean distance as a measure of similarity. Gene-enrichment and functional annotation analysis and pathway analysis for a significant gene list were performed based on Gene Ontology and KEGG pathway analyses.

### 4.8. Hierarchical Clustering

Hierarchical clustering analysis was carried out with complete linkage and Euclidean distance as a measure of similarity to indicate the expression patterns of differentially expressed transcripts which are satisfied with |fold change| ≥ 2 and independent *t*-test raw *p* < 0.05. All data analysis and visualization of differentially expressed genes were conducted using R 3.5.1 (www.r-project.org (accessed on 1 October 2021)).

### 4.9. Whole RNA Extraction and Quantitative Real-Time Reverse Transcription PCR (qRT-PCR)

Total RNA was acquired from patient-derived HCC cells by extraction with the RNeasy Mini Kit (Qiagen, Valencia, CA, USA; 74106) and the One-Step RT-PCR Kit (Qiagen; 204243) following the manufacturer’s protocols. All data were normalized to the expression level of α tubulin. Primers for SERCA1, SERCA2, and SERCA3 are listed in Appendix A.

### 4.10. Cell Culture

The patient-derived HCC cells, YUMC-S-H1, YUMC-R-H1, YUMC-R-H2, YUMC-R-H3, YUMC-R-H4, and YUMC-R-H5 were obtained by tumor cell isolation from the patients and cultured in RPMI-1640 medium with 15% fetal bovine serum (FBS; authenticated by short tandem repeat profiling/karyotyping/isoenzyme analysis).

### 4.11. Cell Viability Assay

Cell viability was measured using the 3-(4,5-Dimethylthiazol-2-yl)-2,5-Diphenyltetrazolium bromide (MTT) assay, cells were seeded in 96-well plates at 7 × 10^3^ cells per well and incubated overnight to achieve 80% confluency. Further details on the protocol are in an existing publication [21]. The assay was performed three times and the viability of the cells was measured as a percentage of the signal observed in vehicle-treated cells and is reported as the means ± standard error of the means.

### 4.12. Cytosolic Free Calcium Measurements by Microspectrofluorimetry

Patient-derived HCC cells were cultured and loaded with Fura-2/AM. HCC cells were perfused with 140 mM NaCl, 5.4 mM KCl, 2 mM CaCl_2_, 1 mM MgCl_2_, 33 mM glucose, and 20 mM HEPES (pH 7.4, adjusted with NaOH, and 320–335 mOsm with sucrose). The cytosolic free calcium level of HCC cells was imaged using a calcium-sensitive fluorescent dye, Fura-2AM. The ratio of Fura-2 fluorescence intensities measured with excitation at 340 nm and 380 nm [F340/F380] was used as a [Ca^2+^]_i_-related signal. Fluorescence intensities (∆F) were normalized to the resting values.

### 4.13. Immunoblot Analysis

The primary antibodies sarco/endoplasmic reticulum calcium ATPase (SERCA, 1:200, Santa Cruz Biotechnology, #271669), C/-EBP homologous protein (CHOP, 1:100, Santa Cruz Biotechnology, #7351), Bcl-2 (1:500, Cell Signaling Technology, Danvers, MA, USA, # 4223S), and β-actin (1:2000, Santa Cruz Biotechnology, CA, USA, #47778) were purchased and maintained overnight at 4 °C. Equal amounts of protein were separated on 8–10% sodium dodecyl sulfate-polyacrylamide gels; the resolved proteins were electro-transferred onto polyvinylidene fluoride membranes (Millipore, Bedford, MA, USA). The membranes were subsequently blocked with 5–10% nonfat milk in TBST for 1 h at room temperature and incubated with appropriate concentrations of primary antibodies overnight at 4 °C. The membranes were then rinsed 3–10 times with TBST and probed with the corresponding secondary antibodies conjugated to horse radish peroxidase (Santa Cruz) at room temperature for 1 h. After rinsing, the blots were developed with ECL reagents (Pierce) and exposed using Kodak X-OMAT AR Film (Eastman Kodak, Rochester, NY, USA) for 1–5 min. Further protocol details are as previously published [75]. 

### 4.14. Cell Cycle Analysis Using Flow Cytometry

Cells were treated with anti-cancer drugs (regorafenib, sorafenib, cisplatin, and gemcitabine) and SERCA inhibitors (thapsigargin, candidates 19 and 23) as either agent alone or combination in RPMI-1640 medium containing 15% FBS for 40 h. The cells were then harvested by trypsinization and fixed in 70% ethanol. Cells were stained for total DNA using PBS containing 40 μg/mL propidium iodide and 100 μg/mL RNase I for 30 min at 37 °C. The cell cycle distribution was then analyzed using the FACSCalibur Flow Cytometer (BD Biosciences, San Jose, CA, USA). The proportions of cells in the sub-G_0_/G_1_, G_0_/G_1_, S, and G_2_/M phases were analyzed using FlowJo v8 for MacOSX (Tree Star, Ashland, OR, USA). This experiment was repeated in triplicate and the results were averaged. Further details on the protocol can be found in our previous article [76].

### 4.15. Pharmacophore- and Docking-Based Sequential Virtual Screening for the Identification of a Novel SERCA Inhibitor

The potential SERCA-binding chemical compounds were screened by evolutionary chemical binding similarity (ECBS), which was built based on classification similarity-learning to prioritize evolutionarily related chemical pairs (ERCPs). By ECBS, chemical pairs are considered as “similar” when their binding targets are identical or evolutionarily related. Among variants of ECBS models, the target-specific ensemble ECBS (TS-ensECBS) model was adapted for the virtual screening owing to the highest test accuracy in our previous study [26]. The TS-ensECBS model was built for SERCA (i.e., the ERCPs are only defined for SERCA and its homologous proteins). The model was then used to calculate chemical binding similarity (ECBS score) between previously known SERCA inhibitors (obtained from DrugBank and BindingDB database) and the virtual chemical library (141,102 chemicals combined from Maybridge and Chembridge screening collection, and DrugBank). The maximum ECBS score assigned for each molecule in the chemical library was considered as a final SERCA-binding score. The output similarity score ranged from 0 to 1, and the scores closer to 1 represent a higher binding probability to SERCA. More details about the ECBS model can be found in our previous work [60].

### 4.16. Patient-Derived Hepatocellular Carcinoma (HCC) Cell Xenograft

Whole experiments were within the guidelines established by the Animal Experiment Committee of Yonsei University. YUMC-R-H1, -H2, -H3, -H4, and -H5 patient-derived HCC cells (4.4 × 10^6^ cells/mouse) were cultured in vitro. The patient-derived HCC cells were then injected subcutaneously into the upper left flank region of female NOD/Shi-scid, IL-2Rγ KOJic (NOG) mice. After 15 days, tumor cell planted mice were randomly grouped (*n* = 10 per group). The mice were then administered with 40 mg/kg regorafenib (p.o.), 80 mg/kg sorafenib (p.o.), 5 mg/kg cisplatin (p.o.), and 40 mg/kg gemcitabine alone, or a combination 20 mg/kg regorafenib (p.o.), 40 mg/kg sorafenib (p.o.), 2.5 mg/kg cisplatin (p.o.), and 20 mg/kg gemcitabine. Candidate 19 and 23 was treated 25 mg/kg (p.o.) alone or combination with regorafenib, sorafenib, cisplatin and gemcitabine. Tumor volume was quantified every two days by calipers. Tumor volume was gauged by the following formula: L × S2/2 (L, longest diameter; S, shortest diameter). Mice were kept under specific pathogen-free conditions. 

### 4.17. Statistical Analysis

For the assay of patient reports, unequivocal fluctuations were expressed as frequency and proportion whereas summary statistics (median, range) were used to report continuous data. Survival curves were generated using the Kaplan–Meier method based on the log-rank test. As this was a retrospective analysis, no formal statistical compari

Sons were performed. Statistical analyses were performed using GraphPad Prism 6.0 software (GraphPad Software, La Jolla, CA, USA), Microsoft Excel (Microsoft Corp, Redmond, WA, USA), and R version 2.17. One-way ANOVA was performed for the multigroup analysis, and a two-tailed Student’s *t*-test was performed for the two-group analysis. Immunohistochemistry results were subjected to one-way analysis of variance, followed by a Bonferroni post hoc test. Values were expressed as mean ± standard error of mean. *p* values < 0.05 were considered statistically significant.

## 5. Conclusions

SERCA activation promoted cytosolic free calcium influx into the ER, prevented cytosolic free calcium overload, and was primarily responsible for cellular resistance to genotoxic stress and apoptosis triggered by anti-cancer drug treatment. Of note, candidates 19 and 23 in combination with anti-cancer drugs could markedly inhibit drug-resistant HCC. Therefore, our findings might represent the tip of an iceberg of numerous signaling pathways involved in the survival of anti-cancer drug-resistant HCC cells. However, the experiments on patient-derived drug-resistant cancer cells were challenging and limited. Nevertheless, these challenges might lead to a breakthrough in and clinical solution to drug-resistant cancer.

## Figures and Tables

**Figure 1 ijms-23-07971-f001:**
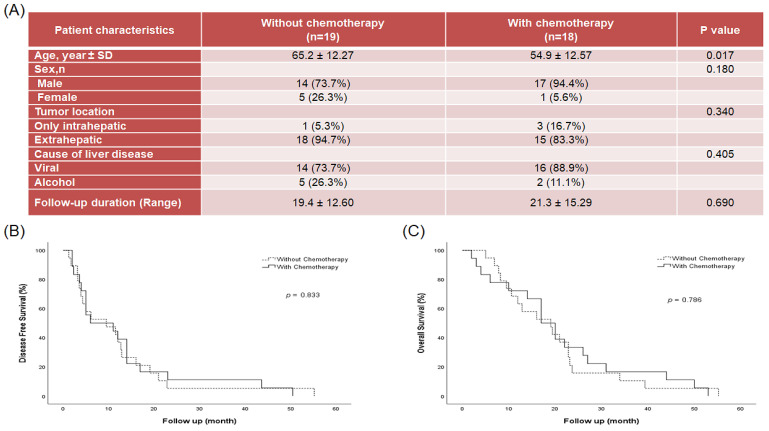
Information on patients with HCC after surgery with or without chemotherapy. (**A**) Patient characteristics and clinical features. (**B**) Disease-free survival rate after surgery with or without chemotherapy. (**C**) Overall survival rate after surgery with or without chemotherapy.

**Figure 2 ijms-23-07971-f002:**
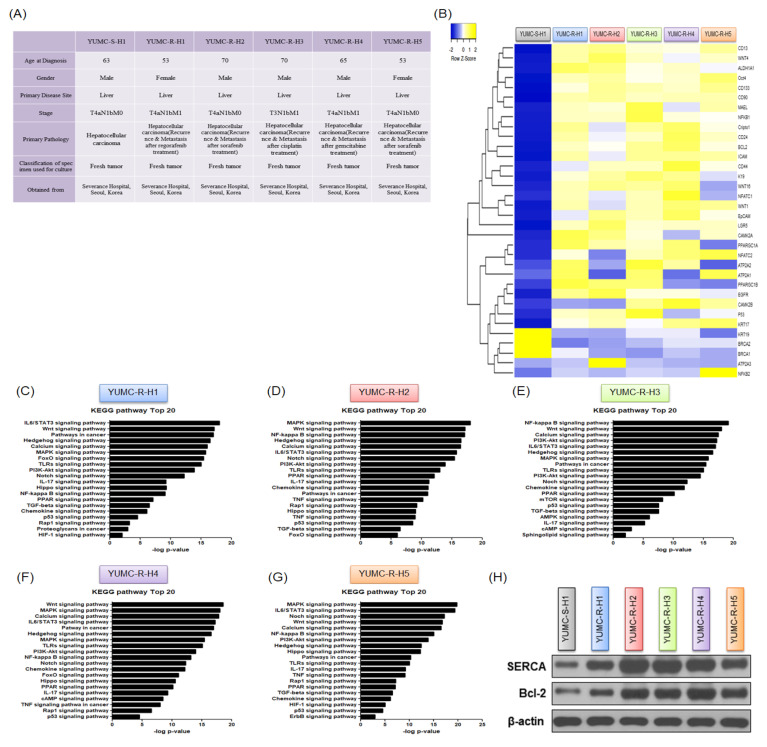
Features of patient−derived HCC cell lines. (**A**) Features of patient-derived subtypes of drug-sensitive and drug-resistant HCC cell lines. (**B**) Hierarchical clustering of annotated genes revealed different gene expression. Differences in gene expression profiles between drug-sensitive and drug-resistant HCC cells. (**C**–**G**) Bar plot showing 20 markedly upregulated pathways in YUMC−R−H1, −H2, −H3, −H4, and -H5 compared with those in YUMC−S−H1. (**H**) Immunoblot analysis of SERCA levels in HCC cells. (**I**–**M**) qRT−PCR and mRNA expression of SERCA family genes of drug-resistant HCC cells compared with those of drug-sensitive HCC cells. ** *p* < 0.01 versus anti-cancer drug-sensitive HCC cells, YUMC-S-H1.

**Figure 3 ijms-23-07971-f003:**
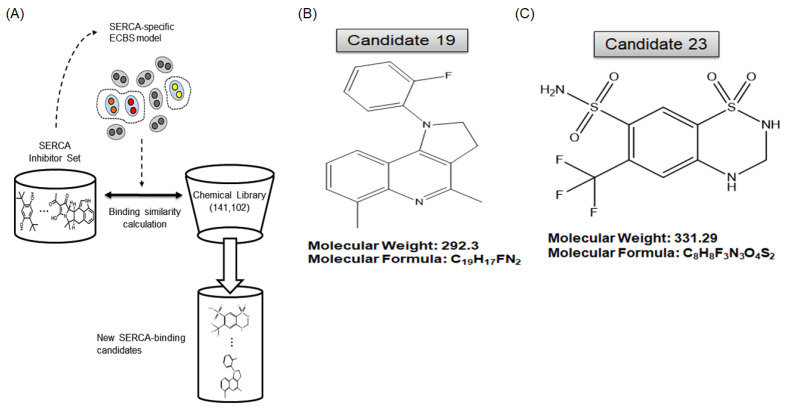
In silico screening using the evolutionary chemical binding similarity (ECBS) method for the discovery of SERCA inhibitors. (**A**) The general categorization of diverse ECBS models is presented with an example. (**B**,**C**) Chemical structures and information of candidates 19 and 23.

**Figure 4 ijms-23-07971-f004:**
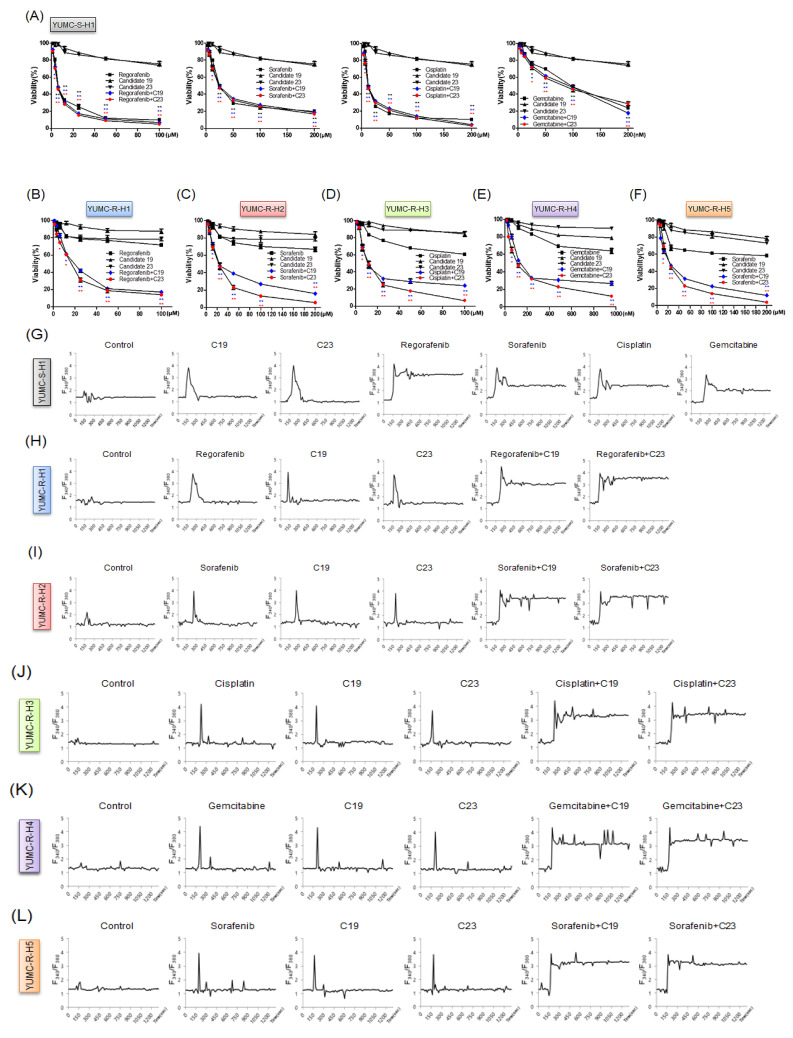
Synergistic anti-cancer effects of candidates 19 and 23 in patient-derived anti-cancer drug-resistant HCC cells. The viability of (**A**) YUMC-S-H1 and (**B**–**F**) YUMC-R-H1, -H2, -H3, -H4, and -H5 treated with regorafenib, sorafenib, cisplatin, or gemcitabine combined with candidates 19 or 23 or with each agent alone. Points indicate the mean percentage of the values of solvent-treated control. All experiments were repeated at least thrice. Data represent the mean ± standard deviation. * *p* < 0.05 and ** *p* < 0.01 versus control. (**G**–**L**) Cytosolic free calcium was measured in patient-derived HCC cells treated with anti-cancer drugs.

**Figure 5 ijms-23-07971-f005:**
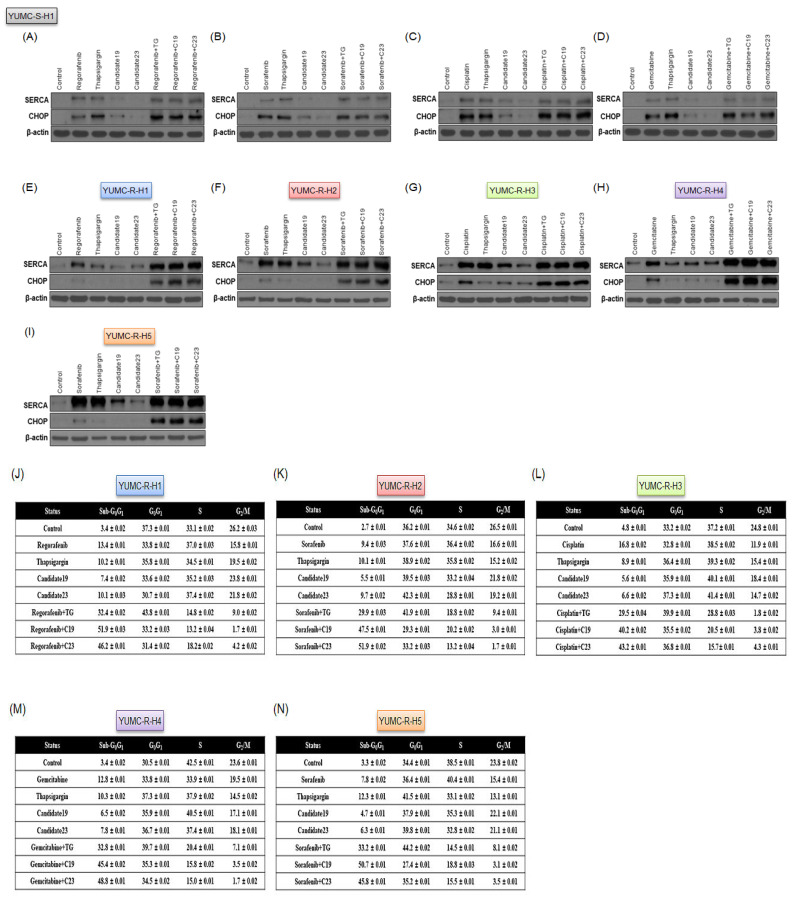
Analysis of the relationship between endoplasmic reticulum (ER) stress and apoptosis by immunoblotting and DNA quantitation. Immunoblot (**A**–**I**) and flow cytometry (**J**–**N**) analyses of patient-derived anti-cancer drug-sensitive and -resistant HCC cells. Immunoblot analysis of CHOP and SERCA in YUMC-S-H1 (**A**–**D**) and YUMC-R-H1, -H2, -H3, -H4, and -H5 (**E**–**I**). (**J**–**N**) Cells were treated with the indicated anti-cancer drug combined with a SERCA inhibitor or with each agent alone, harvested, and stained with propidium iodide before analysis using flow cytometry. (**O**–**S**) Immunoblot analysis of CHOP in drug-resistant HCC cells treated with a combination of an anti-cancer drug, calcium channel blockers, and a SERCA inhibitor or with each agent alone.

**Figure 6 ijms-23-07971-f006:**
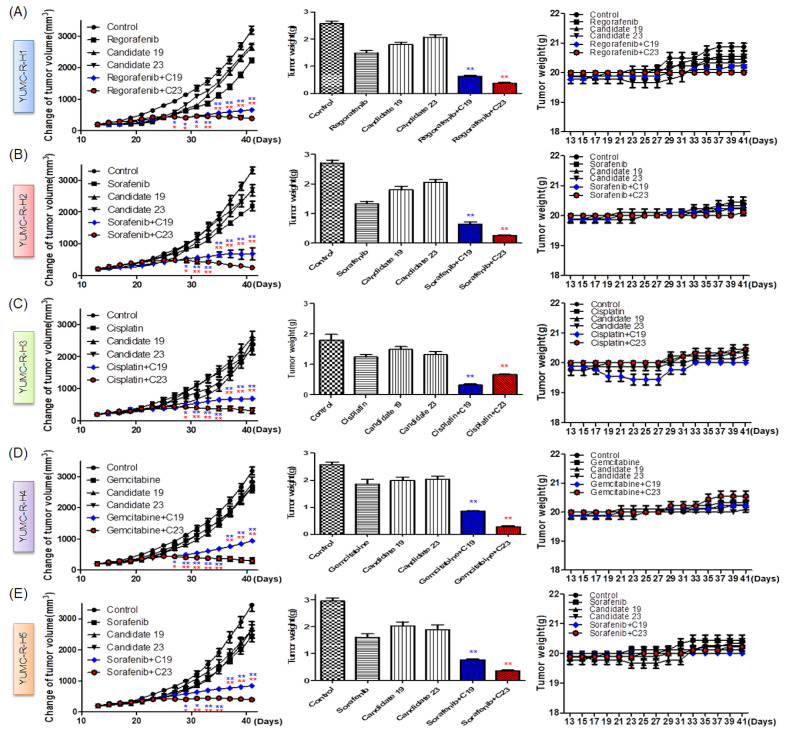
SERCA inhibitors effectively suppress tumor growth in a xenograft model using drug-resistant HCC cells. (**A**) YUHS-R-H1, (**B**) YUHS-R-H2, (**C**) YUHS-R-H3, (**D**) YUHS-R-H4, and (**E**) YUHS-R-H5; changes in relative tumor volumes (**left**), resected tumor weights (**middle**), and body weight (**right**) of mice (each group, *n* = 10). Tumors were established in NOD/Shi-scid, IL-2Rγ KOJic (NOG) mice and animals were treated with regorafenib, sorafenib, cisplatin, or gemcitabine combined with candidates 19 or 23 or with each agent alone. Data represent the mean ± standard error of the mean. * *p* < 0.05 and ** *p* < 0.01 versus control.

**Table 1 ijms-23-07971-t001:** Sample data from patients with hepatocellular carcinoma.

Cells name	PatientsNumber	Age	Sex	First Operation	Second Operation	Third Operation	Fourth Operation	Chemotherapy Regimen before Specimen Was Obtained
YUMC-S-H1	1	63	male	**laparoscopic left lateral sectionectomy ***	□	□	□	none
YUMC-R-H1	2	53	female	left trisectionectomy and caudate segmentectomy with radical resection of common bile duct	**palliative abdominal wall mass excision ***			Sorafenib
YUMC-R-H5	2	53	female	left trisectionectomy and caudate segmentectomy with radical resection of common bile duct	palliative abdominal wall mass excision	**palliative chest wall mass excision ***		sorafenib, regorafenib
YUMC-R-H2	3	70	male	laparoscopic right anterior sectionectomy	laparoscopic aortocaval lymph node dissection	**superior mesenterioc lymph node dissection ***		Sorafenib regorafenib
YUMC-R-H3	3	70	male	laparoscopic right anterior sectionectomy	laparoscopic aortocaval lymph node dissection	superior mesenterioc lymph node dissection	**Multiple lymph node dissection ***	Sorafenib regorafenib Foluorouracil and Cisplatin
YUMC-R-H4	4	63	male	wedge resection of segment 5	right hepatectomy *****	□	□	Gemcitabine and Cisplatin

***** Operation from which the specimens for culture were obtained.

## Data Availability

The data presented in this study are available on reasonable request from the corresponding author.

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
