# Peer review of "Drug Discovery Using Evolutionary Similarities in Chemical Binding to Inhibit Patient-Derived Hepatocellular Carcinoma"

_ijms, 2022, doi:10.3390/ijms23147971_

Round 1
Reviewer 1 Report
Jin Hong Lim and his colleagues evaluated clinical approach to the use of candidate compounds 19 and 23, which are sarco/endoplasmic reticulum calcium ATPase (SERCA) inhibitors, discovered using the evolutionary chemical binding similarity method. . Candidate compounds 19 and 23, were found to be novel SERCA inhibitors which induced significant tumor shrinkage in a xenograft tumor model of anti-cancer drug-resistant patient-derived HCC cells. it is an interesting novel data but the following points should be addressed
1-flow chart of study design is recommended
2-patients samples data in the methodology should be tabulated to be easier to follow
3- more details of mRNA sequencing should be added
Author Response
Reviewer 1
Jin Hong Lim and his colleagues evaluated clinical approach to the use of candidate compounds 19 and 23, which are sarco/endoplasmic reticulum calcium ATPase (SERCA) inhibitors, discovered using the evolutionary chemical binding similarity method. Candidate compounds 19 and 23, were found to be novel SERCA inhibitors which induced significant tumor shrinkage in a xenograft tumor model of anti-cancer drug-resistant patient-derived HCC cells. it is an interesting novel data but the following points should be addressed.
Reply: I don't know how to thank you enough for reviewing our manuscript. I hope you are always healthy and happy!!
- flow chart of study design is recommended
Reply: I agree with you completely and follow your professional opinion. I have made the suggested and included chart of study design in ‘Materials and methods’ and ‘supplementary figure 1’ (see blow).
- patients samples data in the methodology should be tabulated to be easier to follow
Reply: I follow your professional opinion, patient samples data-related table was added in ‘Materials and methods’.
- more details of mRNA sequencing should be added
Reply: I follow your professional opinion. I have made the suggested and included details of mRNA sequencing in Materials and methods (‘2.6. mRNA-seq data’ and ‘2.7. Statistical analysis of gene expression level’).

Reviewer 2 Report
The Paper “Drug discovery to inhibiting patient-derived hepatocellular carcinoma growth using the evolutionary chemical binding similarity” reports important data about HCC.
The content is informative and drug resistance arouses a lot of interest.
Author reports several data about the evaluation of a clinical approach to the use of 19 and 23 compounds.
I suggest minor revision:
Material and methods
2.13. Immunoblot analysis: Authors should provide more data for primary antibodies, dilution etc….).
Results
The molecular weight must be reported in all immunoblot images.
Authors should re-read the whole paper in order to eliminate some repeated concept and errors in sentence construction.
Author Response
Reviewer 2
The Paper “Drug discovery to inhibiting patient-derived hepatocellular carcinoma growth using the evolutionary chemical binding similarity” reports important data about HCC.
The content is informative and drug resistance arouses a lot of interest.
Author reports several data about the evaluation of a clinical approach to the use of 19 and 23 compounds.
Reply: I don't know how to thank you enough for reviewing our manuscript. I hope you are always healthy and happy!!
2.13. Immunoblot analysis: Authors should provide more data for primary antibodies, dilution etc….).
Reply: I agree with you completely and follow your professional opinion. I have made the suggested correction. Corrected sentence were indicated red color. Thank you again for your review.
The molecular weight must be reported in all immunoblot images.
Reply: I agree with you completely and follow your professional opinion. I have made the suggested correction.
Authors should re-read the whole paper in order to eliminate some repeated concept and errors in sentence construction.
Reply: I follow your professional opinion, the repeated concept were deleted and corrected in the whole paper. Thank you again for your review.
